# Impact of Water Regimes and Amendments on Inorganic Arsenic Exposure to Rice

**DOI:** 10.3390/ijerph18094643

**Published:** 2021-04-27

**Authors:** Supriya Majumder, Pabitra Kumar Biswas, Pabitra Banik

**Affiliations:** 1Department of Soil Science and Agricultural Chemistry, Institute of Agriculture, Visva Bharati 731236, Sriniketan, India; majumdersupriya@gmail.com (S.M.); ppabitra07@rediffmail.com (P.K.B.); 2Agricultural and Ecological Research Unit, Indian Statistical Institute, Kolkata 700108, India

**Keywords:** arsenic, water regime, amendments, inorganic arsenic, health risk

## Abstract

Rice-based diet faces an important public health concern due to arsenic (As) accumulation in rice grain, which is toxic to humans. Rice crops are prone to assimilate As due to continuously flooded cultivation. In this study, the objective was to determine how water regimes (flooded and aerobic) in rice cultivation impact total As and inorganic As speciation in rice on the basis of a field-scale trial in the post-monsoon season. Iron and silicon with NPK/organic manure were amended in each regime. We hypothesised that aerobic practice receiving amendments would reduce As uptake in rice grain with a subsequent decrease in accumulation of inorganic As species relative to flooded conditions (control). Continuously flooded conditions enhanced soil As availability by 32% compared to aerobic conditions. Under aerobic conditions, total As concentrations in rice decreased by 62% compared to flooded conditions. Speciation analyses revealed that aerobic conditions significantly reduced (*p* < 0.05) arsenite (68%) and arsenate (61%) accumulation in rice grains. Iron and silicon exhibited significant impact on reducing arsenate and arsenite uptake in rice, respectively. The study indicates that aerobic rice cultivation with minimum use of irrigation water can lead to lower risk of inorganic As exposure to rice relative to flooded practice.

## 1. Introduction

Arsenic (As) is highly toxic among the other trace elements ubiquitously found in the environment. As toxicity is one of the biggest calamities in the world. The source of As in the terrestrial environment is found to be natural as well as anthropogenic [1]. Groundwater contamination through naturally occurring As leads to subsequent mobilisation of As in the soil matrix. It gives rise to the potential impact on groundwater-irrigated paddies considering the source of As input into the food crops [2]. It is crucial to investigate rice As exposure in the As-contaminated rice agroecosystem continuum.

The majority of the world population mostly rely upon a rice-based consistent diet. Among the other cereals, paddy rice is prone to assimilate As more efficiently [3]. Therefore, consumption of rice is a significant As exposure route to humans [4]. The four most common forms of As in rice grain are arsenite (As III), arsenate (As V), dimethylarsinic acid (DMA), and monomethylarsonic acid (MMA); however, the presence of MMA is rare in rice. Arsenite and arsenate are inorganic As species, whereas DMA and MMA are organic As species. Chemical speciation determines the toxicity of As. Inorganic species are toxic to cellular metabolisms [1].

The broad distributions of inorganic As species and their relationship with total As concentration in rice grain have been reported in numerous studies across different countries. A first market basket survey was conducted by [5] on rice varieties originating from Bangladesh. Inorganic As was detected with dominant As species ranging from 60 to 83%. In another study from West Bengal (India), inorganic As contributed to 81.8–88.1% in rice grain [6]. Islam et al. [7] reported the range of inorganic As concentration of 85–97% in rice grain of different cultivars grown in Bangladesh. In another study, the predominant As species detected in rice grains were As (III) (19.8–54.4%) and DMA (45.6–80.2%) for rice grown in As elevated paddy soils from Taiwan [8]. Inorganic As was distributed with wide variability ranging from 0.4 to 96% in Italian-grown rice, and 41 to 97% in the Iberian Peninsula region of Europe, indicating a strong influence of geographic origin [9,10].

As in rice mainly originates from the soil. Dissolution of As from parent material and irrigation water used both are the likely natural source of soil As build-up. Arsenical pesticides used also contribute to elevated soil As concentration [11]. In Bangladesh, the source of irrigation water resulted in increasing As concentration in rice grain up to 1.7 µg g^−1^ during dry season rice cultivation [12]. Pumping of irrigation water mostly extracts groundwater from shallow aquifers that are contaminated with an enormous quantity of dissolved As, imposing efficient rice exposure to As [13].

Soil microorganisms influence on As biogeochemistry affecting the soil redox conditions and the release of As from As-bearing minerals into pore water [14,15]. Traditional flooded cultivation practice mostly triggers enhanced bioavailability of As in rice. Paddy field with anaerobic conditions can mobilise soil-bound As into soil solution through the reductive dissolution of Fe oxyhydroxides, leading to the enhanced prevalence of As (III). In contrast, As (V) predominates in the oxidised paddy field, whose mobilisation is restricted due to strong adsorption to soil mineral constituents [16]. Microbial influence on As speciation in rhizosphere soil by converting inorganic to organic form can regulate the translocation of As from rice roots to edible parts [17,18]. These give rise to critically impact on rice uptake of As species.

Inorganic As is considered as a group “A” carcinogen by USEPA [19]. Long-term exposure to inorganic As has been found to cause cancer and non-carcinogenic health risks such as diabetes, cardiovascular disease, and neurotoxicity [11]. A study showed that the consumption of rice is a significant source of human As intake for the Asian countries in particular [20]. The U.S. Food and Drug Administration proposed the recommendation level of 100 µg·kg^−1^ for inorganic As in rice and rice-based baby foods. However, rice cultivation has received more precise attention in many parts of the world, having a higher level of inorganic As content (>200 µg·kg^−1^; WHO limiting value). Therefore, specific soil management strategies should identify to minimise the risk of inorganic As accumulation in rice grain.

Multiple studies indicate that rice cultivation without maintaining continuous flooding may impact reduce As accumulation in rice grain. As speciation in rice grain can be affected by soil drying [16]. The severity of soil drying to water potential less than −33 k Pa at the surface soil (0–15 cm) decreased total grain As concentration in rice grain [21]. In another study, Carrijo et al. [22] reported that drying severity at the booting or heading stage instead of panicle initiation causes a decrease in inorganic As accumulation in rice grain. Li et al. [23] proposed an alternate wetting and drying (AWD) method for sustained rice cultivation. They observed an overall 14–61% decrease in inorganic As concentration in rice grain for AWD treatment as compared to continuously flooding. Despite the apparent significance of water management practice, the incorporation of soil amendments has the potential implications for minimising rice uptake of As. Iron- and silicon-based amendments are of great importance, having been documented in a number of studies [24,25,26,27]. Iron-based amendments can effectively adsorb As and decrease the mobility of As in soil. Silicon amendments inhibit As uptake in rice by the competitive inhibition at the plant root surface. Organic fertilisers as additive can reduce mobility of As in soil due to the formation of coordinate linkage between phenolic -OH and -COOH groups of organic matter and As anions [28,29]. However, the combined studies of water management practice incorporating the amendments were mostly investigated under controlled greenhouse conditions [24,27]. Aerobic rice cultivation through field-scale framing practice with minimum irrigation requirement is still needed to monitor rice grain As bioaccumulation as well as speciation in As-contaminated environments.

This study investigated how the aerobic water regime receiving iron and silicon amendments impacted total As bioaccumulation and inorganic As speciation in rice grain on the basis of a field-scale trial in the post-monsoon season. We hypothesised that the aerobic regime with specific management practice would reduce the quantity of total and inorganic As species in rice grain compared to continuously flooded regime. This study is able to explore future policy related to rice cultivation with minimum use of groundwater irrigation in As-contaminated regions.

## 2. Materials and Methods

### 2.1. Field Trials

Field trials were conducted at farmer’s plot (23° N, 88° E) of Deganga block, West Bengal (India), during the post-monsoon period (January to April), with *Satabdi* rice cultivar. The experiment was carried out in a split-plot design with 14 treatment combinations. The two levels of water regime used: continuously flooded (control) and aerobic, were included as the main plots. Seven levels of amendment were included in the subplots with a size of 5 m × 3 m. The levels of each factor contained three replications [30]. Rice seedlings of 30 days were transplanted. During the experiment, the aerobic regime was maintained by unflooded irrigation (1–2 cm depth) until grain maturity. As concentration in irrigation water was 198 µg·L^−1^. In the flooded water regime, the field was kept flooded to a depth of 5–10 cm up to grain maturity.

Seven levels of amendment regime as subplot treatment were added to each water regime. The source of amendment included NPK (N, P, K fertilisers with recommended dose), silicon (silica gel), iron (ferrous sulphate), and organic manure from vermicompost (VC) and farmyard manure (FYM), individually on each. The levels included control (no amendment); silicon + NPK; iron + NPK; silicon + FYM; iron + FYM; silicon + VC; iron + VC. The FYM and VC were applied at the rate of 16 t·ha^−1^ and 8 t·ha^−1^, respectively. Ferrous sulphate and silica gel were applied at 75 kg·ha^−1^ and 500kg·ha^−1^ and, respectively. The recommended doses of N, P, K fertilisers were 120 kg N (urea), 60 kg P_2_O_5_, and 60 kg K_2_O·ha^−1^, respectively.

### 2.2. Soil Sample Collection and Measurement of Soil Properties

The initial soil properties before the study are represented in Table 1. Composite soil samples were collected (0–15 cm) at harvest from each amended plot. The samples were air-dried and sieved (<2 mm) before analysis. Soil samples were analysed by the standard procedure as described for texture (international pipette method), soil pH [31], and organic carbon [32]. Amorphous Fe oxide was extracted by ammonium oxalate buffer solution (pH 3.25) [31]. Dithionite citrate bicarbonate (DCB)-extracted Fe was measured by the procedure of Jackson [31]. For analyses of available silicon, the soil sample was extracted with 0.5 M acetic acid (1:10) [33].

### 2.3. Determination of Soil-Extractable As

Soil-extractable As was determined by 0.01 M CaCl_2_ following the procedure of Houba et al. [34].

### 2.4. Plant Sample Collection

Rice grain samples were collected from each plot at harvest. The grain samples were polished and preserved in a plastic bag for As estimation.

### 2.5. Digestion Procedure

Rice grain sample of 0.5 g was transferred in a Teflon vessel and digested with HNO_3_ (69%, Emplura, Merck) and H_2_O_2_ (30%, Merck) at 4:1 on a hot plate at 120 °C for 5 h. The dilution of the digested solution was made up to 10 mL using deionised water and filtered and stored in a plastic tube for As analysis.

### 2.6. Determination of Inorganic As Species in Rice Grain

A total of 0.5 g of powdered rice grain was extracted with 5 mL of 0.28 M HNO_3_ using a heating block at 95 °C for 90 min, as described by Huang et al. [35]. The digests were centrifuged at 5858× *g* for 15 min and passed through 0.45 µm syringe filter paper, and then were finally stored at 4 °C in the dark for speciation analysis. All the analyses were performed with in 72 h. Sodium meta-arsenite and sodium arsenate dibasic heptahydrate (sigma) were included as standards for speciation analysis. The speciation extracts and matrix match standards were quantified by an HPLC system coupled with ICP MS (Perkin Elmer Nexion 300, USA). Hamilton PRP-X100 anion exchange column was used in the HPLC system. The mobile phase was 20 mM NH_4_H_2_PO_4_, and pH was adjusted to 5.6 with NH_4_OH.

### 2.7. Instruments and Quality Control

Microwave plasma atomic emission spectrometry (MP-AES; Agilent 4210, Agilent, USA) was used for total As analysis. Certified reference material of rice and blank was analysed for As to remove batch-specific error (in triplicate). Details of quality control data are represented in Appendix A.

### 2.8. Statistical Analyses

Statistical design of field experiment was followed according to the methodology of [30]. Data were statistically analysed in R statistical package *Agricolae* (v. 1.3–1). ANOVA was performed to test the statistical significance (*p* = 0.05). Correlation analysis was performed using cor package in R.

## 3. Results

### 3.1. Paddy Soil As

Soil extractable As concentrations were significantly impacted by the type of water regimes (Table 2). As concentrations in paddy soil under aerobic conditions reduced by 32% compared to flooded conditions. Among the soil amendments, treatments receiving ferrous sulphate significantly reduced soil available As concentrations compared to control (*p* < 0.05) (Table 2). Silica gel resulted in a non-significant difference compared to control. There was a significant interaction effect of water regime and amendment on available As concentrations (for further details please check Appendix A).

### 3.2. Total As in Rice

In aerobic conditions, total As concentrations in rice grain decreased by 62% compared to flooded conditions (Figure 1A). Among the soil amendments, the grain As concentrations reduced by 18–23% with silica gel application compared to control, while ferrous sulphate applied with VC and FYM detected the lowest significant grain As accumulation compared to other treatments (*p* < 0.05) (Figure 1B). There was a significant interaction effect of water regime and soil amendment, while the lowest As accumulation was detected in ferrous sulphate addition with VC/FYM combinations under aerobic conditions (Figure 1C).

### 3.3. Grain Inorganic As Speciation

Percentage distributions of inorganic As species in polished rice are represented in Figure 2A. Relative distributions of As (III) and As (V) in terms of water regime and soil amendment treatments are shown in Figure 2B,C, respectively. As (III) was the dominant species detected in polished rice, regardless of water regime and soil amendment treatments. As (III) and As (V) concentrations under aerobic condition were reduced by 68% and 61%, respectively, compared to flooded condition. Relative to amendments, ferrous sulphate and silica gel with either NPK or organic manures significantly reduced As (III) accumulation in rice grain compared to control (*p* < 0.05). The concentration of As (V) was significantly reduced by ferrous sulphate applications compared to control and silica gel treatments. The results strongly suggest that aerobic conditions and amendment applications effectively reduced inorganic As species concentrations in rice grain.

### 3.4. Correlations of As Concentrations in Rice Grains and Soil Chemical Properties

The concentrations of total As, arsenite, arsenate, and sum of inorganic As species in rice grains showed significant positive correlations with soil-extractable As (Table 3). It clearly indicates that soil available As was closely related to increasing uptake of As in rice grain. The amount of amorphous iron and DCB-extracted iron in soil was significantly negatively correlated to rice grain As concentration, regardless of As species. We also noted a significant negative correlation between the amount of available silicon in soil and As (III) concentration in rice grain, which could indicate silicon mediated antagonistic impact on reducing As (III) uptake in rice. Amorphous iron, DCB-extracted iron, and available Si concentrations are shown in Appendix A.

## 4. Discussion

### 4.1. Paddy Soil As Response to Water Regimes and Amendments

Water regimes greatly impact the redox reactions in paddy soils since As is redox-sensitive. The redox reactions in paddy soils strictly regulate the biogeochemistry of As. Oxic soil conditions under aerobic regime influenced the reduction of As mobilisation. In contrast, anaerobic conditions can promote As solubility in paddy soils, leading to increased As availability through the mechanism of reductive dissolution of Fe oxyhydroxides, which accounts for releasing co-precipitated As to the soil solution. Numerous studies have reported that rice cultivation under aerobic conditions reduces As concentration in paddy soil and soil pore water compared to anaerobic conditions [36,37]. Relative to amendments, availability of As was decreased in paddy soils with ferrous sulphate treatments compared to control. Iron is considered as providing the most essential adsorption sites for soil As, maintaining availability and mobility of As in soil [38]. As (V) in the oxic soil conditions can specifically adsorb to iron oxides [39,40]. It could positively impact reducing As availability in paddy soils.

In contrast, available As concentration tended to be higher while silica gel was incorporated. Competition between silicic acid and arsenous acid can release As from soil solid phase to the solution site by ligand exchange reactions [41,42]. It may occur in the flooded conditions to a greater extent. Application of organic manure as an additive might influence As sorption through complex formation with As, leading to a decrease in As release in soil solution [43,44].

### 4.2. Inorganic As response in Rice to Water Regimes and Amendments

Our study demonstrated a dramatic influence of water regime on total As concentration and inorganic As speciation in polished rice grain. Results indicate that the aerobic system plays a functional role to minimise As accumulation in rice grain. Consistent with As availability in the soil, As bioavailability to rice under flooded treatment was much more enhanced compared to aerobic conditions. Apart from the abiotic factors (pH, organic carbon, redox potential, metal hydroxide), biotransformation of As in paddy soil by microorganisms can enzymatically regulate accumulation and speciation of As in paddy rice [18]. As oxidising and reducing bacteria often coexist in the rice rhizosphere and soil biota, which are sensitive to the redox conditions. Under anaerobic conditions, As (V)-reducing and Fe (III)-reducing bacteria could enhance the dissolution of As from Fe oxyhydroxide into the soil pore water. The consequent reduction of As (V) to As (III) causes the desorption of As into the solution phase from the surface of Fe oxyhydroxides [16], which could promote the higher incorporation of As (III) into rice. On the other hand, oxygenated root under aerobic conditions can develop the formation of co-precipitated As with Fe minerals by the activities of As (III)-oxidising and Fe (II)-oxidising bacteria, leading to the limited dissolution of As into soil solution [18,45]. Therefore, aerobic conditions could substantially decrease the bioavailability of As species in rice grain. Numerous studies have documented the trends of reduced As bioavailability in rice grain, maintaining the soil under aerobic conditions on the basis of greenhouse experiments [24,37]. In contrast to As, rice exposure to cadmium (Cd) is an emerging concern because Cd is regarded as a well-known food contaminant. Previous studies determined that Cd and As have reverse responses to water management [46,47]. Aerobic conditions decreased As bioavailability in rice grain while increased Cd concentration was detected. Flooded soil conditions with higher pH and sulphide content can lead to immobilisation of Cd compared to aerobic soils [48]. However, Cd concentration was not detected under the present study of experimental conditions.

Rice grains contained inorganic As in almost the majority of total As concentration, ranging from 67% to 87%. The extent of total As content in rice grain was similarly translated into inorganic As concentration on the basis of the type of water regimes. We found that there was higher inorganic As concentration in rice grain under flooded conditions than in aerobic conditions, which was associated with large differences in total As concentration in rice grain between the type of water regimes. These findings suggest that soil flooding increased the inorganic As concentrations in rice grain, demonstrating a logarithmic increasing relationship with soil-available As (Figure 3). It clearly indicates that inorganic As was readily taken up by plants under the flooded system compared to aerobic conditions. The higher proportions of inorganic As were in agreement with earlier investigations of rice grown in Asian countries [5,7,49].

As (III) was the dominant inorganic species in both of the water regimes. Rice plants usually take up As (III) from the soil with a higher magnitude compared to As (V) [50]. Accumulation of As (III) might be enhanced by the direct absorption of pore water in the rhizosphere system and the consequent transport into grain under flooded treatment compared to aerobic condition. Regression analyses of total As against inorganic As species across the treatments of amendment and water regime explained the satisfactory level of variation with the significantly positive relationship having a higher slope of As (III) (Figure 4). This consequence suggests the higher translocation of As (III) in rice compared to As (V).

Total As and inorganic As content in rice grain were also varied with the impact of silicon fertilisation. Rice is known as a strong silicon accumulator compared to other cereal crops. Silicon is a beneficial element. Silicon is conducive to reducing As uptake by rice because of the competitive reaction between As (III) and silicon at the soil exchange site. Due to the sharing of similar transport pathway, silicon inhibits As (III) uptake by rice. As in the form of As (III) is readily assimilated in rice plant through a subclass of aquaporins, which are known as nodulin 2-like intrinsic proteins, NIPs. These proteins contain some silicon transporters which secrete As (III) from the roots and transport As (III) into the xylem [51]. As (III) uptake in rice favours the silicon transport pathway, sharing a competitive reaction between As (III) and silicon in soil. As (III) is a competitive analogue of silicic acid (H_4_SiO_4_) [52], sharing an antagonistic relationship. Aquaporin *LSi1* mediates H_4_SiO_4_ influx into the roots and transports As (III). *LSi2* is another silicon transporter which expresses at the inner side of the root plasma membrane, releasing As (III) into the soil [52]. The competitive reaction between silicon and As (III) in soil solution could inhibit subsequent excess As (III) transport to the grain. It was confirmed by significant negative correlations between paddy soil available silicon and As (III) concentration in rice grain. Some studies have documented that silicon addition decreased total and inorganic As concentration in rice grain [24,27].

Iron application inhibited the As uptake in rice, which was probably due to the iron plaque formation. Rice roots can release oxygen to the rhizosphere. It has the potentiality to oxidise iron by forming precipitation of Fe oxy hydroxide (FeOOH) at the root surface. It can sequester As and therefore minimise As uptake by rice [23]. As (V) has a much higher affinity to iron compared to As (III) [16]. It is assumed that As (III)-oxidising bacteria in the rhizosphere under aerobic conditions could impact higher As (V) binding on iron material and iron plaque. It could substantially reduce As (V) uptake and As bioavailability in rice [18]. Iron amendments increased the concentration of DCB-extracted iron in the soil (Appendix A). It can restrict the release of As from the soil solid phase due to co-precipitation, therefore minimising total As uptake in rice. Results showed that iron application significantly reduced inorganic As concentration in rice grain also. It was also supported by significant negative correlations between the amounts of amorphous iron and DCB-extracted iron in the soil and inorganic As species concentration in rice grain.

## 5. Conclusions

This study provides an important field-scale approach to rice grown with the variation of cultivation practices in As stress environment. The results indicate that aerobic rice cultivation with a special management practice has the efficacy to mitigate As accumulation in rice grains. Oxic soil under aerobic conditions resulted in reduced As transfer to grain. Importantly, the aerobic practice affected the grain As speciation, which resulted in decreasing inorganic As accumulation. Amendments application led to a positive impact on reducing total As uptake in rice and inorganic As species. Overall, the aerobic regime, combined with the application of amendments, reduced the risk of inorganic As accumulation in rice grain when compared to flooded practice. The work corroborates a fundamental approach on rice cultivation technique in As-contaminated environments. Moreover, aerobic rice cultivation is preferred in terms of reducing As in paddy rice; however, this should not be applicable if rice is grown in Cd-contaminated soil. Future field-scale investigations should be focused to optimise irrigation management to obtain concomitantly reduced uptake of As and Cd in rice. 

## Figures and Tables

**Figure 1 ijerph-18-04643-f001:**
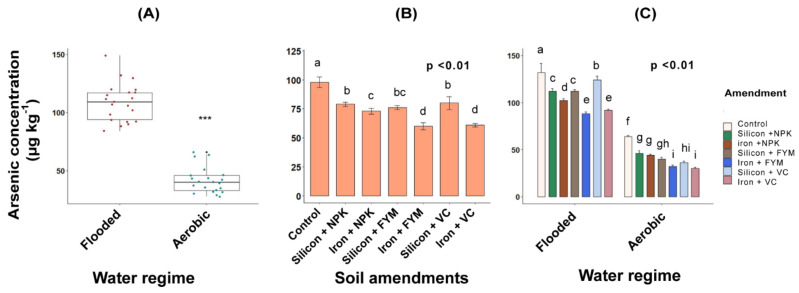
Total As concentration in rice grain in terms of water regime variation (**A**); soil amendment application (**B**); and interaction between water regime and amendment (**C**). Different letters indicate significant differences between treatments from Fisher’s LSD test (*p* < 0.05); *** indicates significance level at *p* < 0.001.

**Figure 2 ijerph-18-04643-f002:**
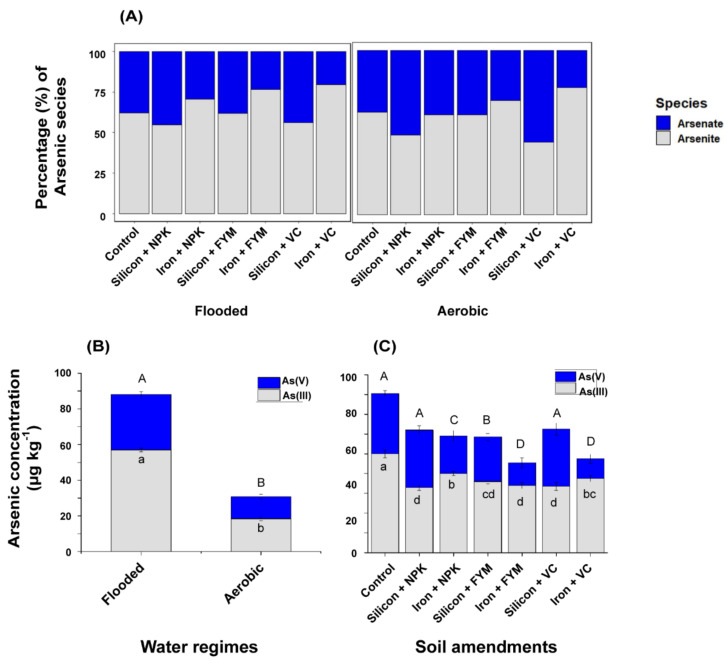
Percentage distribution of inorganic As species in rice grain subject to the effect of soil amendment and water regime (**A**). Arsenite and arsenate concentrations in rice grain in terms of water regime variation (**B**) and soil amendment application (**C**). Error bars represent standard error of means. Different lowercase and uppercase letters indicate significant differences between treatments from Fisher’s LSD test (*p* < 0.05).

**Figure 3 ijerph-18-04643-f003:**
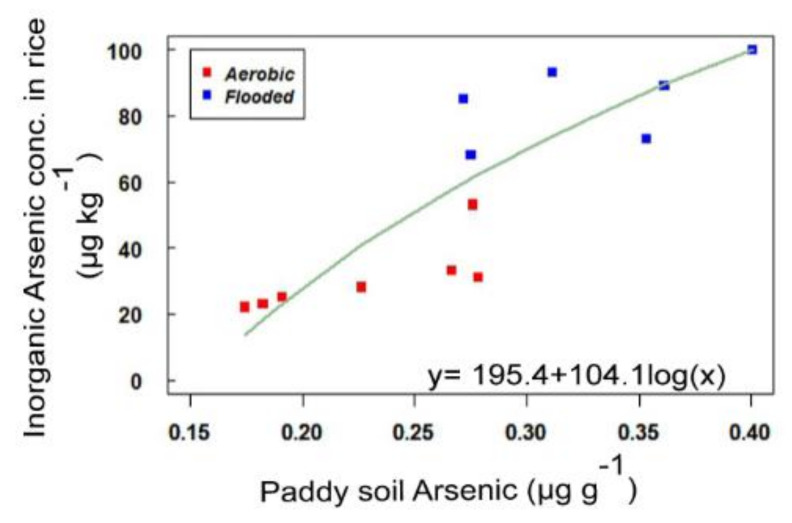
Relationship between paddy soil As and inorganic As concentrations in rice grain.

**Figure 4 ijerph-18-04643-f004:**
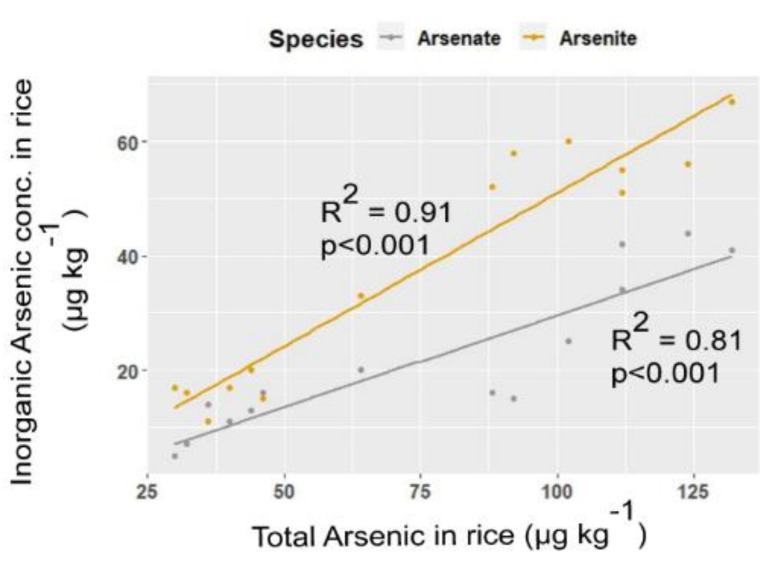
Correlations between total As concentration and inorganic As species in rice grain.

**Table 1 ijerph-18-04643-t001:** Initial physicochemical properties of experimental soil.

Parameters	Values
Sand (%)	25.2
Silt (%)	34.9
Clay (%)	39.9
Texture	Clay loam
pH	7.59
Organic carbon (%)	0.81
Amorphous Fe (g·kg^−1^) DCB-extracted Fe (g·kg^−1^)	1.92 5.43
Total As (mg·kg^−1^)	11.62

**Table 2 ijerph-18-04643-t002:** Effect of water regimes and amendments on available As concentrations in paddy soils.

Factors		Soil As (mg·kg^−1^)
**Water regime**		
Flooded		0.33 ± 0.06 a
Aerobic		0.21 ± 0.08 b
**Soil amendments**		
Control		0.31 ± 0.035 a
Silicon + NPK		0.29 ± 0.045 ab
Iron + NPK		0.27 ± 0.051 c
Silicon + FYM		0.29 ± 0.028 ab
Iron + FYM		0.23 ± 0.019 d
Silicon + VC		0.30 ± 0.058 ab
Iron + VC		0.26 ± 0.028 c
*p*-Value	Water regime	0.0005
	Soil amendments	<0.0001
	Water regime × soil amendments	<0.0001

Values followed by different letters represent significant difference from Fisher’s LSD test (*p* < 0.05).

**Table 3 ijerph-18-04643-t003:** Correlation coefficients between As species in rice grain and soil chemical properties.

Rice Grain	Soil Properties			
	Soil-Extractable As	Available Si	Amorphous Fe	DCB-Extracted Fe
Arsenite	0.80 ***	−0.64 ***	−0.86 ***	−61 **
Arsenate	0.79 ***	−0.38 *	−0.88 ***	−86 ***
Sum of As species	0.84 ***	−0.55 **	−0.92 ***	−83 ***
Total As	0.87 ***	−0.65 **	−0.92 ***	−64 **

*, **, and *** indicate significance levels at *p* < 0.05, *p* < 0.01, and *p* < 0.001, respectively.

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
