# Peer review of "Impact of Water Regimes and Amendments on Inorganic Arsenic Exposure to Rice"

_ijerph, 2021, doi:10.3390/ijerph18094643_

Round 1
Reviewer 1 Report
Majumder et al. reported the effects of water regimes and fertilizers amendments (both minerals and organics) on arsenic (As) bioaccumulation in paddy rice under both flooded and aerobic conditions. They conducted farmer’s field based experiment and investigated if water in combination of fertilizer amendments can reduce As accumulation in rice. They also performed As speciation study to explore the percentage of inorganic As in grain and the non-carcinogenic health effect. The results are very interesting and this could certainly insight some kind of alternative way of paddy cultivation in arsenic-enriched soil to prevent As-accumulation in rice grain. The flow of the work is satisfactory, but anyhow, by any means the focus or objective of the study is missing. The authors should have to explain why they prefer this kind of paddy cultivation practice particularly using arsenic contaminated soil or water? Plenty of research reports are available under aerobic or flooded conditions, but why this study is important and differ from others?
Lines 41-42: MMA is rare in rice. Authors should provide percentage of different As species detected in rice grain along with appropriate references.
Line 50: please check the reference whether Roychowdhury articles provided the inorganic As percentage as 81.8-88.1%.
Introduction needs to be revised as there is no flow at this moment. Authors should summarize the current water management practices on arsenic reduction in paddy rice. It was also not clear why authors have selected water regimes and different fertilizers amendments-the rational for choosing amendments should be clearly stated.
Section 2.6; No reference was used for the method of As speciation analysis.
Although rice CRM was used for As speciation for QA/QC issue, no organic As concentration was provided. Table S2 is okay as authors have provided inorganic As concentrations whereas Table S3, both AsIII and AsV individually were provided which is dubious as 0.28M nitric acid was used for As speciation extraction. There is a huge chance to redox transformation of inorganic As species and hence sum of inorganic As was reported in literature, not the individual levels of AsIII and AsV.
As Fe amendments were used, this has formed the iron plaque at the root surface which hindered As transportation to the grain. No reason or investigation was provided why As in grain was less when both water regime and fertilizer amendments were used. The mechanistic reason should be investigated. Usually DCB extracted Fe should be measured to find out the reason of As reduction in grain.
Line 255: what is SRI?
Line 402: The MTDI value of 2.1 was withdrawn in 2011, authors still use the same which is also dubious. Authors should use the current benchmark values by various organizations to compare their daily dietary intake and health risk.
In general, the figure captions used in this manuscript are not self-explanatory. Therefore, the authors are suggested to include some additional information in each figure caption to make it sound and easily understandable to the readers.
Reviewer 2 Report
The manuscript "Impact of water regimes and amendments on inorganic Arsenic exposure to rice: Assessment of potential human health risk" is appropriate for the journal and the subject is relevant for a large part of the world. The document is well structured, concise, but very effective. Tables are adequate. The figures appear misplaced in the printed version that I have consulted. I think they can be improved in their aesthetics and require a more professional treatment. Table 1 of the supplementary material could go in the main body of the work, either as a table or described with text. Soil characteristics are important to evaluate the behavior of arsenic.
The writing is very correct. For the rest, it is a fairly concise work but that provides information of interest, so that, once the problems with the graphics have been corrected, I believe that it deserves publication.
Reviewer 3 Report
It is known already from different countries that rice cultivated under aerobic conditions has lower arsenic content, but also that the cadmium content increases under such conditions; see e.g. the study below with the arsenic-in-rice expert Professor Meharg. Therefore the proposed manuscript provides nothing new. On the contrary, it may lead some people to the impression that there is an easy solution to the problem with arsenic in rice. Cadmium is an equally severe problem as arsenic. Thus, I would not recommend publish a manuscript that preserves outdated knowledge. Sprinkler irrigation of rice fields reduces grain arsenic but enhances cadmium. Moreno-Jiménez E, Meharg AA, Smolders E, Manzano R, Becerra D, Sánchez-Llerena J, Albarrán Á, López-Piñero A. Sci Total Environ. 2014 Jul 1;485-486:468-473. doi: 10.1016/j.scitotenv.2014.03.106. Epub 2014 Apr 16. PMID: 24742557 Previous studies have demonstrated that rice cultivated under flooded conditions has higher concentrations of arsenic (As) but lower cadmium (Cd) compared to rice grown in unsaturated soils. To validate such effects over long terms under Mediterranean …Author Response
Please see the attachment.

Round 2
Reviewer 1 Report
I belive that authors have improved the manuscript significantly base don the reviewers comments. However, I also agree with reviewer 3 and hence requested authors to modify the conclusion that aoerobic rice cultivation is preferred to reduce As in paddy rice, however, this should not be applicable if rice is grown in Cd-contaminated soil.
Reviewer 3 Report
The authors´ argument about the potential novelty of the performed field studies on arsenic accumulation in rice may be accepted. However, the response about cadmium," Cadmium is a special issue related to rice. However, cadmium was not detected under the study of experimental conditions. Therefore, we completely disagree with this particular point.", is not acceptable in my opinion. The public health concern about environmental exposure to cadmium, of which rice is an important source, is growing rapidly. I propose that the authors mention the additional problem with cadmium in rice, and what is known about cadmium accumulation in rice grown under anaerobic conditions in the discussion. That cadmium was not measured may be listed as a limitation of the study. In fact, I miss such a section in the discussion. Obviously, the manuscript will benefit markedly by a more complete and holistic discussion of the pros and cons about different cultivation conditions for rice.
Concerning the public health discussion on arsenic, I will additionally focus on section "4.3 Dietary intake and non-carcinogenic risk", which is difficult to understand. What is meant by non-carcinogenic risk? Acute or chronic exposure? If the latter, what time frame is considered? How about cancer risks? The section starts with several specific statements, without explanation or references. Some examples:
-“Consumption of rice with higher content of inorganic As can adversely impact the human health risk." What is meant by higher content of inorganic As? Please, specify the type of health risk considered.
-"Rice originated from India mostly contains a higher proportion of inorganic As." Higher than what? -"Populations are likely to be at risk of high As exposure from the rice-based diet." Which populations? What risk? What is meant by high arsenic?
-"Our study indicates that rice cultivation practices were the major contributor to impact health risk parameters related to the As exposure." However, the exposure to arsenic was not assesses.
-"We determined dietary intake of inorganic As exposure and contribution to MTDI from the rice consumption under flooded and aerobic conditions from Equation 1 (Table 4)." No, the authors did not determine the dietary intake of As. An estimation was made of the intake of arsenic from rice, not from other foods included in the diet. What is MTDI? Importantly, the health risks of arsenic cannot be based on just on type of rice. Different types are consumed over seasons and in a longer time perspective. Also, a single field experiment in one specific area and type of soil be used for such an estimation? What are the uncertainties?
-"On average, the contribution of flooded rice to dietary intake of inorganic As was higher than that of aerobic rice." But that is obvious as the concentration of arsenic in the flooded rice was higher than the rice cultivated under anaerobic conditions. How much did the higher paddy soil concentrations of arsenic (Figure 3) contribute to the higher rice arsenic concentration?
-“We noted that dietary intake values were below the WHO’s updated MTDI value of 3 μg kg-1 BW day-1 for inorganic As (determined from epidemiological studies considering a range of assumption of total dietary exposure to inorganic As from drinking water and food) [52,56].” The reference 52 is not a WHO reference, reference 56 does not work. Please, provide more information.
-“HI was calculated to measure the susceptibility of the local population to non-carcinogenic health impact from the consumption of rice (Equation 2). According to the guideline of USEPA, HI >1 indicates potential risk [57].” This is very difficult to understand. Please, clarify in detail.
Tables should be possible to read without searching for explanations in the text. Table 4 does not. What is HI? What is MTDI?
Considering all these uncertain statements, I would recommend deleting the whole section 4.3 “Dietary intake and non-carcinogenic risk". It has actually very little to do with this single field experiment.
Related not only to this section, it is unclear how the concentration of inorganic arsenic in rice was determined. Please clarify. Concerning the speciation (2.6. Determination of inorganic As species in rice grain), how can AsIII and AsV in rice be determined after the powdered rice is extracted with a strong oxidizing acid like nitric acid?
